# FPGA-Based Reconfigurable Convolutional Neural Network Accelerator Using Sparse and Convolutional Optimization

Kavitha Malali Vishveshwarappa Gowda [1], Sowmya Madhavan [2], Stefano Rinaldi [3,*], Parameshachari Bidare Divakarachari [4] and Anitha Atmakur [5]

1   Department of Electronics and Communication Engineering, Gopalan College of Engineering and Management, Bengaluru 560048, Karnataka, India; kavishanthagiri@gmail.com
2   Department of Electronics and Communication Engineering, Nitte Meenakshi Institute of Technology, Bangalore 560064, Karnataka, India; sowmya.madhavan@nmit.ac.in
3   Department of Information Engineering, University of Brescia, via Branze 38, 25123 Brescia, Italy
4   Department of Telecommunication Engineering, GSSS Institute of Engineering and Technology for Women, Mysuru 570016, Karnataka, India; paramesh@gsss.edu.in
5   Department of Electronics and Communication Engineering, CVR College of Engineering, Hyderabad 501510, Andhra Pradesh, India; a.anitha@cvr.ac.in
*   Correspondence: stefano.rinaldi@unibs.it

**Abstract:** Nowadays, the data flow architecture is considered as a general solution for the acceleration of a deep neural network (DNN) because of its higher parallelism. However, the conventional DNN accelerator offers only a restricted flexibility for diverse network models. In order to overcome this, a reconfigurable convolutional neural network (RCNN) accelerator, i.e., one of the DNN, is required to be developed over the field-programmable gate array (FPGA) platform. In this paper, the sparse optimization of weight (SOW) and convolutional optimization (CO) are proposed to improve the performances of the RCNN accelerator. The combination of SOW and CO is used to optimize the feature map and weight sizes of the RCNN accelerator; therefore, the hardware resources consumed by this RCNN are minimized in FPGA. The performances of RCNN-SOW-CO are analyzed by means of feature map size, weight size, sparseness of the input feature map (IFM), weight parameter proportion, block random access memory (BRAM), digital signal processing (DSP) elements, look-up tables (LUTs), slices, delay, power, and accuracy. An existing architectures OIDSCNN, LP-CNN, and DPR-NN are used to justify efficiency of the RCNN-SOW-CO. The LUT of RCNN-SOW-CO with Alexnet designed in the Zynq-7020 is 5150, which is less than the OIDSCNN and DPR-NN.

**Keywords:** block random access memory; convolutional optimization; field-programmable gate array; reconfigurable convolutional neural network; sparse optimization of weight

## 1. Introduction

Artificial intelligence (AI) is generally a primeval field of computer science and it is extensive worldwide dealing with all the parts of imitating cognitive functions for real-world issue resolving and creating systems which study and think like humans. Hence, it is referred to as machine intelligence instead of human intelligence. This AI is utilized in the integration of computer science and cognitive science. The practical accomplishments in machine learning increases the interest in the field of AI [1]. Examination and application directions of AI technology are learning intelligence, behavior intelligence, thinking intelligence, perception intelligence, and so on [2]. In general, neural networks are used in spacecraft control, vehicle control, pattern recognition, robotics, military equipment, analysis and decision making in the Internet of Things systems, drone control, health care, and so on [3–6]. Specifically, the CNN is one of the modern AI approaches which generates a complex feature when it is processed with a huge model and sufficient training dataset. Therefore, the features from the CNN provide better performance than the typical handcrafted features [7].

The specialized coprocessors comprising application-specific integrated circuits (ASICs), graphics processing units (GPUs), and FPGA use the natural parallelization and offer higher data throughput. The deep neural network is mainly considered in this computing revolution because of the higher parallelizability and general computational requirements [8]. CNN generally includes the intensive multiplication and accumulation operations. These operations are performed sequentially using general-purpose processors that result in low efficiency. The GPU offers the Giga to Tera FLOPs per seconds, but it suffers a high energy cost. FPGA generally uses one order of magnitude less power when compared to GPUs and, also, it provides significant speed improvement compared to the CPUs [9–11]. An effective way to implement the neural network is to utilize the FPGA, due to their effective parallel computing, reconfigurability, and robust flexibility. However, the FPGA also has some limitations, such as implementation complexity, circuit reprogramming, high cost, lack of machine learning libraries, and implementation complexity [12–15]. The proposed SOW and CO optimizations are utilized individually in some of the conventional FPGA applications. However, this research applies SOW and CO techniques to attain advanced quality of performances, which are explained briefly in the following sections.

The contribution of this paper is mentioned as follows:

- This research exploits SOW and CO to decrease the memory print by separating the feature map into minor pieces and storing them in FPGA's high-throughput on-chip memory. This diminishes the weight sizes of RCNN accelerator and accomplishes even higher performances.
- The RCNN accelerator is developed by the FPGA along with the SOW and CO for minimizing the hardware resources.
- The data reuse approach used in the SOW helps to minimize the power consumption. Moreover, a considerable amount of calculation is decreased using the sparse matrix operations.
- Execution speed of the RCNN accelerator is improved by using the loop unrolling approach of SOW, whereas this loop unrolling also minimizes the computing resources.

The remaining paper is arranged as follows: the related work about the existing CNN architectures are described in Section 2. The problems from the related work are specified along with the solutions given by the proposed research in Section 3. Section 4 provides the detailed explanation about the RCNN-SOW-CO architecture. The performance and comparative analysis of the RCNN-SOW-CO are given in Section 5. Lastly, the conclusion is made in Section 6.

## 2. Related Work

The related work about the existing CNN architectures created over the FPGA are given as follows:

Pang et al. [16] presented an end-to-end FPGA-based accelerator to perform an effective operation of fine-grained pruned CNNs. Here, the load imbalances were created after the fine-grained pruning. The accelerator's load imbalances and internal buffer misalignments were resolved by developing a group pruning algorithm with group sparse regularization (GSR). The sparse processing elements design and on and off chip buffers scheduling are used to optimize the accelerator. However, the loss of accuracy was out of control when there was an increment in the number of pruned weights.

Mo et al. [17] developed the deep neural network over the small-scale FPGA to perform the odor identification. The odor identification with depthwise separable CNN (OIDSCNN) was developed for minimizing the parameters and accelerating the hardware design. According to the quantization method, namely the saturation-flooring KL divergence approach was used to design the OI-DSCNN over a Zynq-7020 SoC chip. The optimization in the parameters was used to achieve higher speed. However, the OIDSCNN required a high amount of DSP resources.

Li et al. [18] implemented the AlphaGo policy network and, also, effective hardware architectures were developed for accelerating the deep CNN (DCNNs). The policy network

was implemented to perform the sample actions in the game of Go. The policy network was adjusted to achieve the accurate goal of winning games by combining the reinforcement learning with supervised learning. The designed accelerator was fit in various FPGAs that provided the balance among the hardware resources and processing speed. Due to full utilization of on-chip resources and the parallelism of FPGA, the developed DCNN consumed a high amount of hardware resources.

Vestias [19] designed a hardware-oriented pruning of CNN over the FPGA. In general, the pruning was a model optimization approach which prunes the links among the layers for minimizing the number of weights and operations. The sparsity was introduced in the weights to minimize the computational efficiency of the conventional pipelined architectures. Subsequently, the block pruning was used on the pipelined data path to eliminate the sparsity issue created by the pruning. However, the accuracy of CNN was high only in low-density FPGAs.

Abd El-Maksoud et al. [20] developed the low-power dedicated CNN (LP-CNN) hardware accelerator according to the GoogLeNet CNN. The size of the memory was minimized by applying the weights pruning and quantization. The CNN was processed layer by layer because of the timesharing/pipelined structure-based accelerator. The developed CNN accelerator was used only in an on-chip memory to store the weights and activations. Moreover, the shifting operations were used instead of the multiplications. The accuracy of the LP-CNN was affected because of a huge number of spare weights.

Zhao [21] presented the light music online system by designing the FPGA and CNN. Important tasks performed by the cell neural organization architecture are initiation capacity, ordinariness, pooling, and convolution. However, a huge amount of hardware resources were required to design the CNN-based light music online system.

You and Wu [22] developed a software/hardware co-optimized reconfigurable sparse CNN (RSNN) over the FPGA. The sparse convolution data flow was developed with simple control logic, which used the element-vector multiplication. Here, the software-based load-balance-aware pruning approach was developed for balancing the computation load over dissimilar processing units. The DSP utilization of RSNN was increased because of the simultaneous execution of two 16-bit fixed-point MACs.

Irmak et al. [23] presented an architecture of a high-performance and flexible neural network (NN) accelerator over the FPGA. In this work, dynamic partial reconfiguration (DPR) was developed for realizing the various NN accelerators by updating only a portion of FPGA design. A different processing element (PE) with identical interfaces was designed for computing the layers in various NN. However, the developed NN failed to perform the reconfiguration in FPGA, which resulted in high power consumption.

## 3. Problem Statement

The problems found from related work are provided along with the solutions given by the proposed research.

The accuracy of the overall system is degraded because of the restricted pruning space [16]. For an effective system, the amount of hardware resources is required to be less to minimize the DSP consumption and to improve the speed. However, a high amount of DSP resources is required in OIDSCNN [17]. Moreover, high power consumption is caused because of the NN without any reconfiguration in FPGA [23].

**Solution:** In this research, the reconfigurable convolutional neural network is developed over the FPGA. The design of the RCNN accelerator with the SOW and CO is used to minimize the feature map and weight sizes. Moreover, the RCNN-SOW-CO minimizes the number of calculations according to the optimization approach. This helps to minimize the amount of hardware resources and, accordingly, it increases the speed.

## 4. Overview of Architecture

Figure 1 shows the data-flow-based reconfigurable architecture. Different from the existing structures, the configuration register is introduced, followed by all the accelerator's

hardware modules configured in the RCNN accelerator. The architecture is reconfigured with the configuration register according to the configuration instructions saved in double data rate (DDR). Next, the parameters of weight and image are transmitted to the buffer of weight and image. The PE array streams calculate the outcomes into the special function buffer because it comprises the static random-access memory (SRAM) banks, which are connected in parallel. SRAM cell contains four NMOS transistors and two poly-load resistors. Two NMOS transistors are referred to as pass-transistors. These two transistors take their gates integrated with the word line and attach the cell to the columns. The other two NMOS transistors are connected to pull-downs of the flip-flop inverters. Due to this type of arrangement, integrated SRAM provides faster access to data and can be used for a computer's cache memory. Moreover, the special function buffer has the special functional layers, such as pooling, batch normalization (BN), and activation. These special functional layers minimize data access among the on-chip buffer and DDR. The suggested architecture provides suitable transferability to diverse RCNN models because of its reconfigurable PE array, which can be modified to adapt to numerous filter sizes of the systems. In the meantime, a reconfigurable on-chip buffer procedure is improved while considering the projected model, which completely depends on the restriction ratio property of diverse layers. Furthermore, the RCNN accelerator improves its flexibility by manipulating the sparsity features of the input feature map.

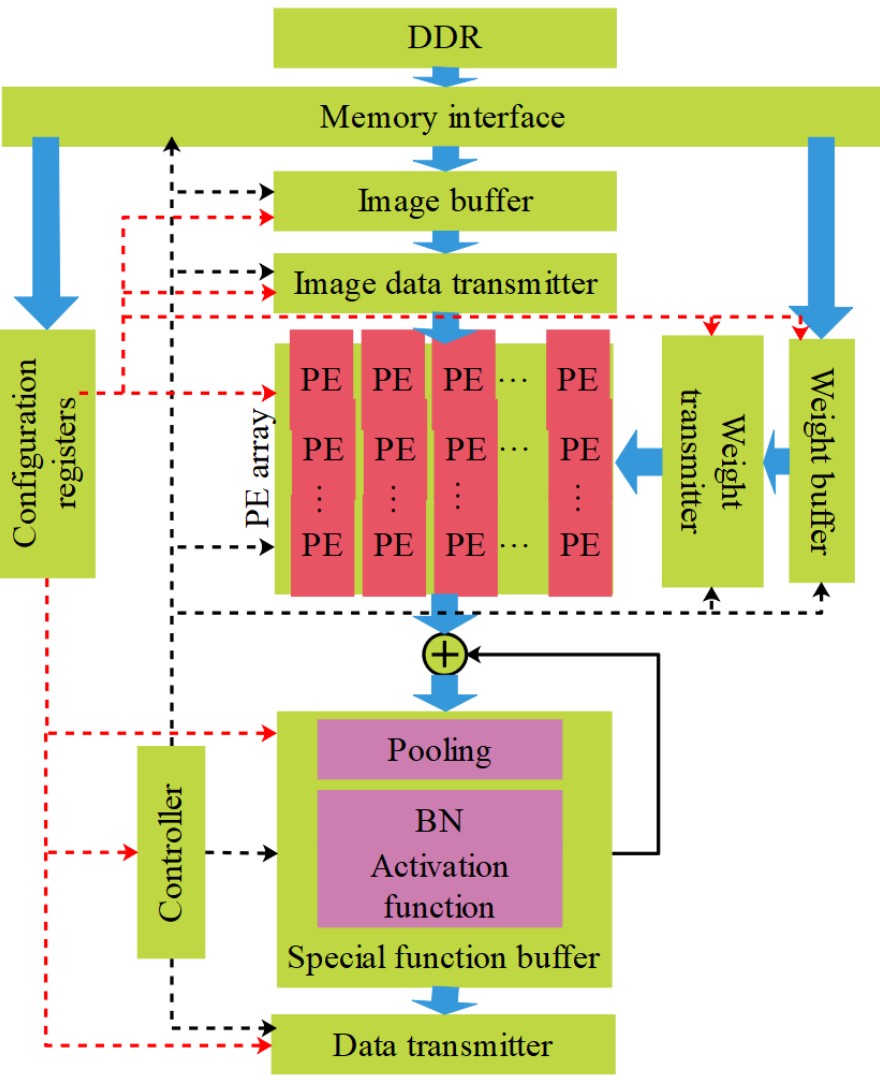

**Figure 1.** Data-flow-based reconfigurable architecture.

### 4.1. Architecture of Reconfigurable PE Used for Convolution

A spatial 2D PE array is implemented to accomplish the tradeoff among the flexibility, throughput, and complexity, whereas the designed spatial 2D PE array is an essential component of the accelerate solution. The spatial 2D structure works better with the complex network structures that have various kernel sizes. Figure 2 displays the architecture of reconfigurable PE.

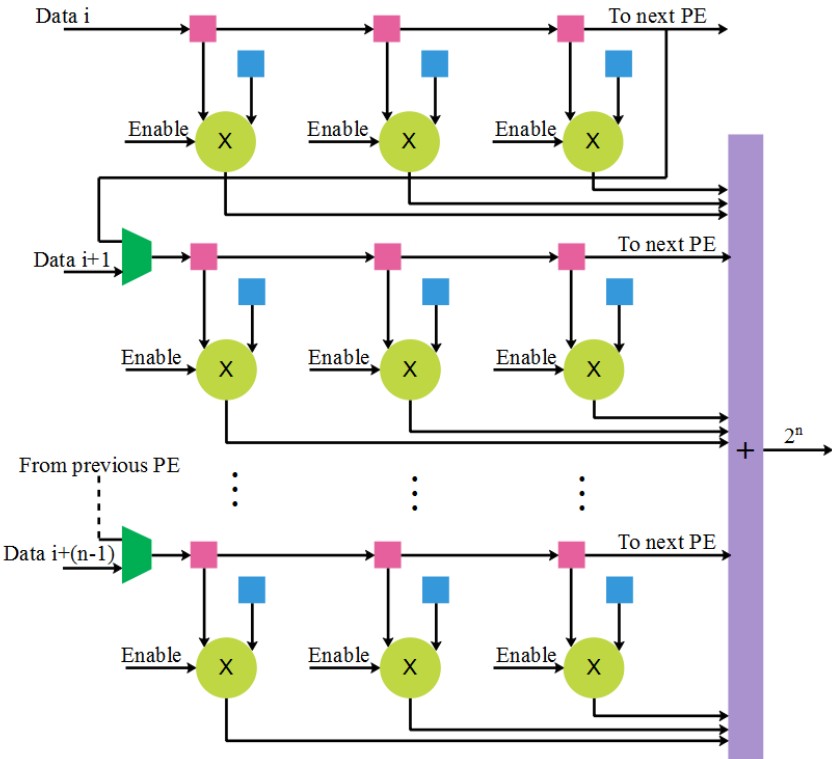

**Figure 2.** Architecture of reconfigurable PE.

The conventional PE design has one multiplier, but the reconfigurable PE has eight multipliers for realizing the convolution operations of $1 \times 1$ or $3 \times 3$, which are the foremost operations in the conventional DNN. Here, eight image registers and eight weight registers are connected to the multipliers. In this design, the image registers are utilized as shift registers and send the image data among the PE. The work status of the PE is decided by the enable signals connected to the multipliers, which helps to minimize the energy consumption. For the inactivated multipliers, the outputs are fixed as zero. The designed reconfigurable PE is used to accomplish the $1 \times 1$ or $3 \times 3$ convolution operations. The accelerator deals with large filter sizes, such as $5 \times 5$ or $7 \times 7$, according to the designed PE array. Moreover, the operation of $11 \times 11$ is also performed by using 16 PEs. The specific configuration of $5 \times 5$ and $7 \times 7$ is shown in Figures 3 and 4, respectively. The usage of multipliers for various kernel sizes is given in the Table 1.

**Table 1.** Peak multiplier utilization for various kernel sizes.

| Kernel Size | Peak Utilization |
|-------------|------------------|
| $1 \times 1$ | 98.3% |
| $3 \times 3$ | 98.1% |
| $5 \times 5$ | 63.1% |
| $7 \times 7$ | 69.12% |
| $11 \times 11$ | 74.01% |

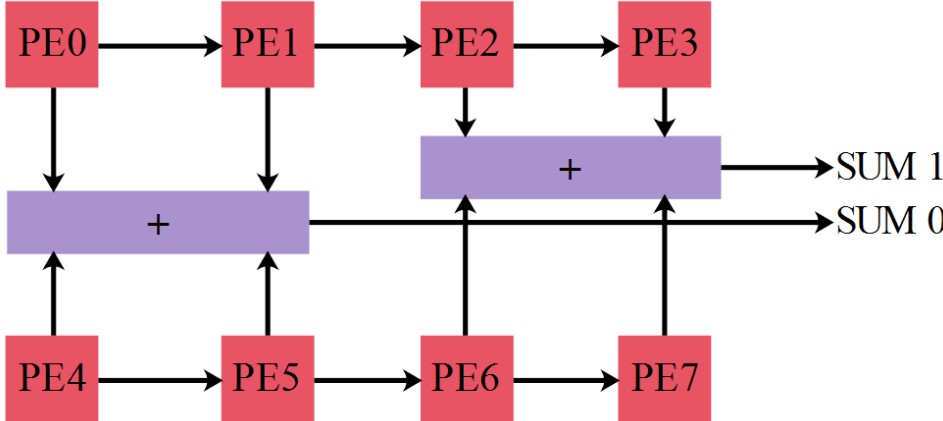

**Figure 3.** Convolutional operation of $5 \times 5$.

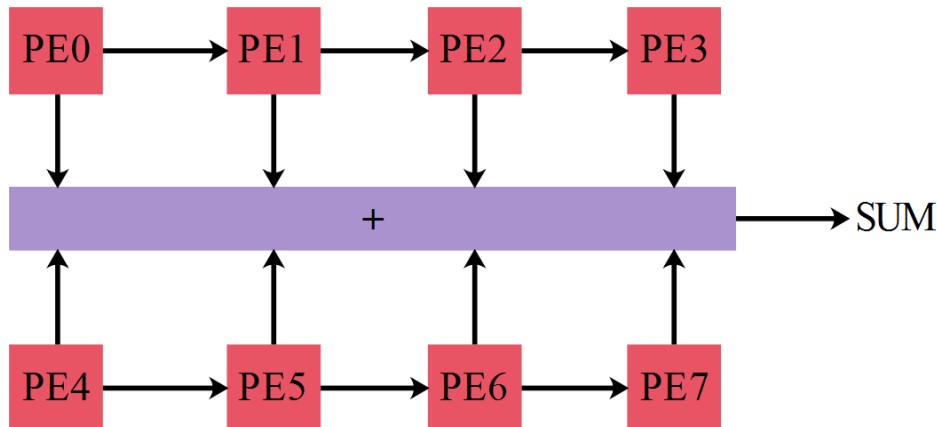

**Figure 4.** Convolutional operation of $7 \times 7$.

The rows of the image are used again for reducing the on-chip data movement, which lessens the power consumption. The reuse approach for the filter size of $3 \times 3$ and a stride of 1 is shown in the Figure 5. Therefore, this reuse approach is used to decrease the usage of SRAM banks.

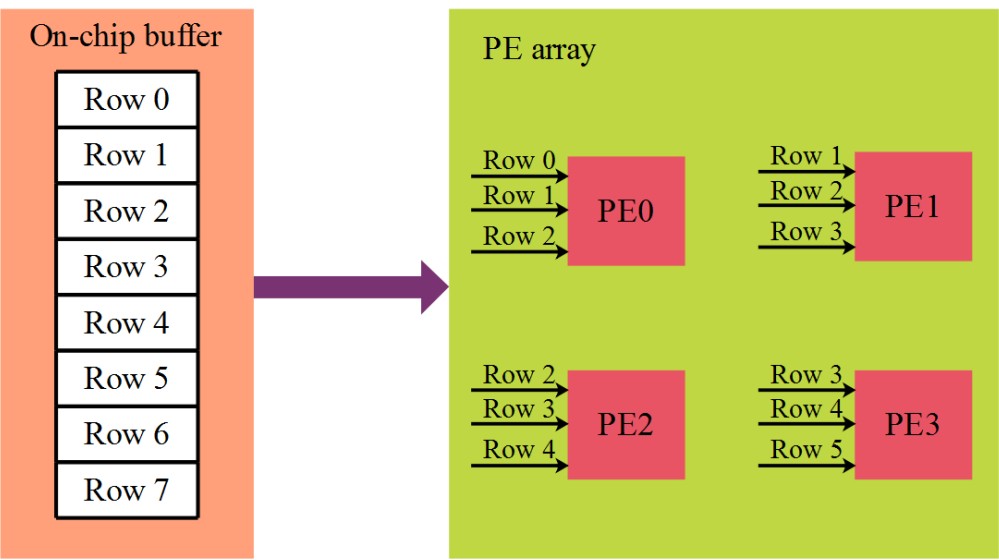

**Figure 5.** Row data reuse strategy.

In this RCNN accelerator, the fully connected (FC) layers are used as the special convolutional layers by using the padding of 0, $1 \times 1$ filter, $1 \times 1$ IFM, and stride of 1. The FC layer has a huge number of weight parameters where the technique used to calculate the weight parameter is explained in the following section.

### 4.2. Sparseness Optimization for Weight

The resources of on-chip memory are inadequate embedded design. A huge amount of energy is utilized for the data access among the on-chip and external memory. Therefore, the data reuse approaches are used in the RCNN accelerator to minimize the power. There are two ideal reuse approaches that are used in the certain layer, such as saving all IFM data on chip and storing all weight parameters on chip. An amount of weight parameter and image information highly differs between various convolutional layers. The approach used to calculate the weight parameter is described as follows.

The spare weight RCNN accelerator is introduced that is considered as applicable for the hardware implementation. Consider the sparse weight has $-w, +w$, and zero. The sparse weight RCNN accelerator contains hidden weights $w^{(hid)}$ while training on the GPU. From the hidden weight, the sparse weight $w^{(t)}$ is defined, which is expressed in Equation (1):

$$w^{(t)} = \begin{cases} 0 & \left| w^{(hid)} \right| \leq \rho \\ w^{(hid)} & Otherwise \end{cases} \tag{1}$$

where the threshold to differentiate the zero weight and a non-zero one is represented as $\rho$.

An example of sparse convolution operation is shown in Figure 6. Moreover, the weight value is always taken as $-w$ or $+w$ for the baseline CNN. Therefore, there is no probability of disconnection between the neurons. The weight 0 state defines the disconnections at the sparse weight RCNN accelerator. The sparse weight RCNN's matrix representation is a sparse one; hence, the operations of sparse matrix is applied for reducing the number of calculations.

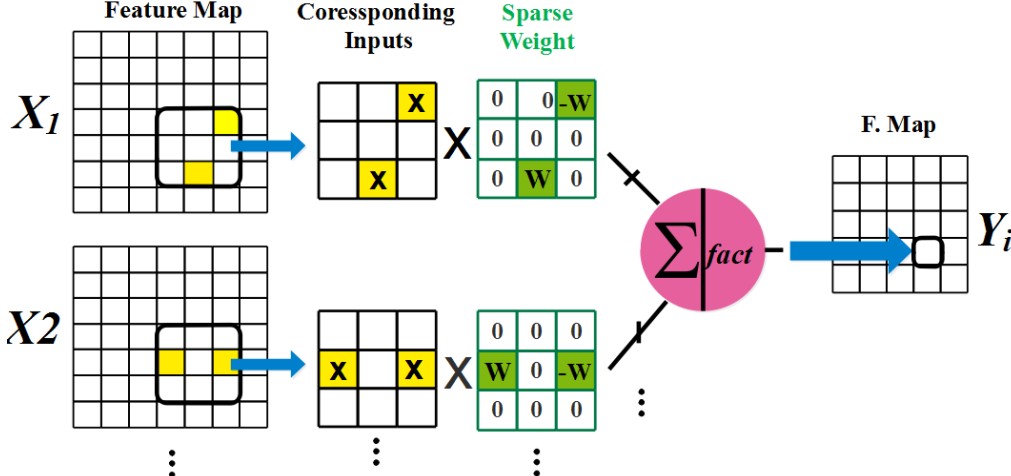

**Figure 6.** Example of sparse convolution operation.

The operation of sparse weight convolutional is realized using the zero-weight skip calculation. Since the address in respect to the non-zero weight is stored, when the pretrained RCNN accelerator includes a zero-weight, consequently, the typical CNN with zero-weight skip one is used to accomplish the sparse weight convolutional operation. The developed convolutional execution needs $L$ words, where an amount of non-zero weights is represented as $L$. Fewer memory accesses are required by the RCNN accelerator with zero weights; however, this RCNN with sparse weight is faster than the 2D calculation.

An indirect memory access is used to perform the zero-skip calculation, whereas the indirect memory access for sparse convolution is shown in Figure 7. Initially, this memory access simultaneously reads the non-zero weight and respective address. Next, this memory access identifies the address for the respective input. Moreover, memory access reads the respective one, followed by it accomplishing the multiply accumulation (MAC) operation. The activation function, i.e., ReLU, is applied by replicating the aforementioned operations for all the non-zero weights in the kernel.

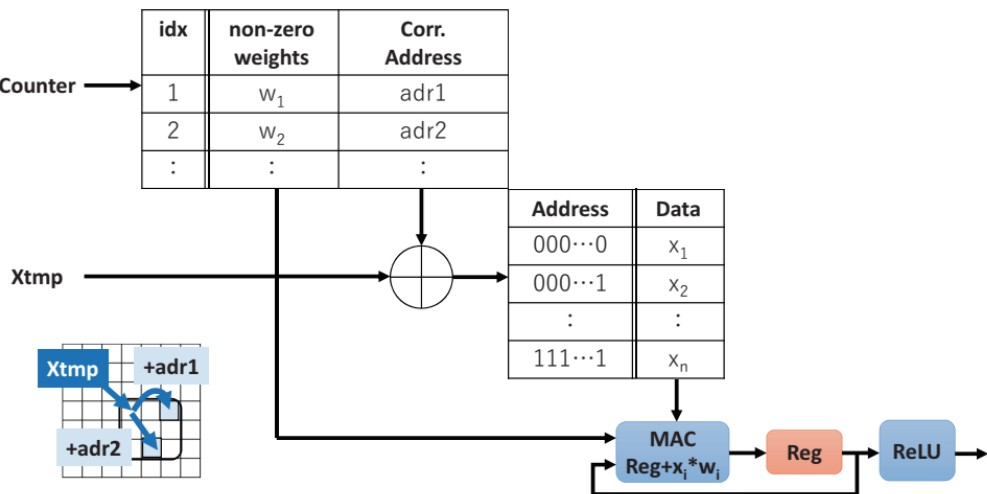

**Figure 7.** Indirect memory access operation.

### 4.3. Convolutional Optimization

The convolution optimization of the convolution (CONV) layer is used for enhancing the performance density. From the Roofline model, the performance density of FPGA accelerator is formulated under definite hardware resource conditions, as shown in Equation (2):

$$
\begin{aligned}
Performance\ bound &= \frac{Total\ number\ of\ operations}{Execution\ cycles} \\
&= \frac{f_{CONV}^{Time} + f_{Pooling}^{Time} + f_{others}^{Time}}{\frac{M}{T_m} \times \frac{N}{T_n} \times \frac{R}{T_r} \times \frac{C}{T_c} \times (T_r \times T_c \times K^2 + p)} \\
&\approx \frac{f_{CONV}^{Time}}{\frac{M}{T_m} \times \frac{N}{T_n} \times R \times C \times K^2} \\
&\approx \frac{g\left(R \times C \times K^2 \times C_{in} \times C_{out}\right)}{\frac{M}{T_m} \times \frac{N}{T_n} \times R \times C \times K^2}
\end{aligned}
$$
$$
0 < T_m \times T_n \times (DSP_{Mul} + DSP_{Add}) < (\#of\ DSP)
$$

where $N$, $M$, $R$ and $C$ denote the layer of input, output, row and column feature maps, respectively; $K$ represents the kernel; tile sizes of the input, output, row and column are $T_m$, $T_n$, $T_r$ and $T_c$, and $p = Pipeline\ depth - 1$; the time complexities of the convolutional and pooling layer are $f_{CONV}^{Time}$ and $f_{Pooling}^{Time}$, respectively. A number of multiply–add operations are used to estimate the time complexity of a certain layer at the RCNN accelerator. Moreover, a number of input channels and convolution kernels in the CONV layer are denoted as $C_{in}$ and $C_{out}$.

More than 90% of operation in the RCNN accelerator is occupied by the convolution operations; therefore, $f_{CONV}^{Time}$ is higher than the sum of $f_{Pooling}^{Time}$ and $f_{others}^{Time}$. Moreover, the tile sizes $T_r = T_c = 1$, whereas the tile sizes of $T_m$ and $T_n$ are variable. Equation (1) shows that the bottleneck of performance is defined only by $T_m$ and $T_n$, which are highly limited by resources of on-chip DSP. The methods used in this convolutional optimization are defined below.

### 4.3.1. Loop Unrolling

The parallelism among CONV kernels is used in the loop unrolling approach to accomplish the parallel execution of many CONV executions. In CONV, the parallel pipeline multiplication and addition are accomplished by partially expanding the two dimensions $M$ and $N$. Here, each layer consists of $N$ IFMs. The pixel blocks ($T_n$) in the identical location and related $T_n$ weights are acquired from autonomous IFMs. Hence, the IFMs require a time of $N/T_n$ to read and compute. The calculation of loop unrolling according to the $N/T_n$ is used to mitigate the resource wastage. The parallel multiplication units $T_m \times T_n$ are used to multiply the input pixel blocks of $T_n$ for performing the multiplication operations, and addition trees of $T_m$ return the product addition output in the output buffer. The local parallel structure is realized by the loop unrolling, as shown in Figure 8. This loop unrolling is used to maximize the speed and concurrently increases the performance for each calculating resource.

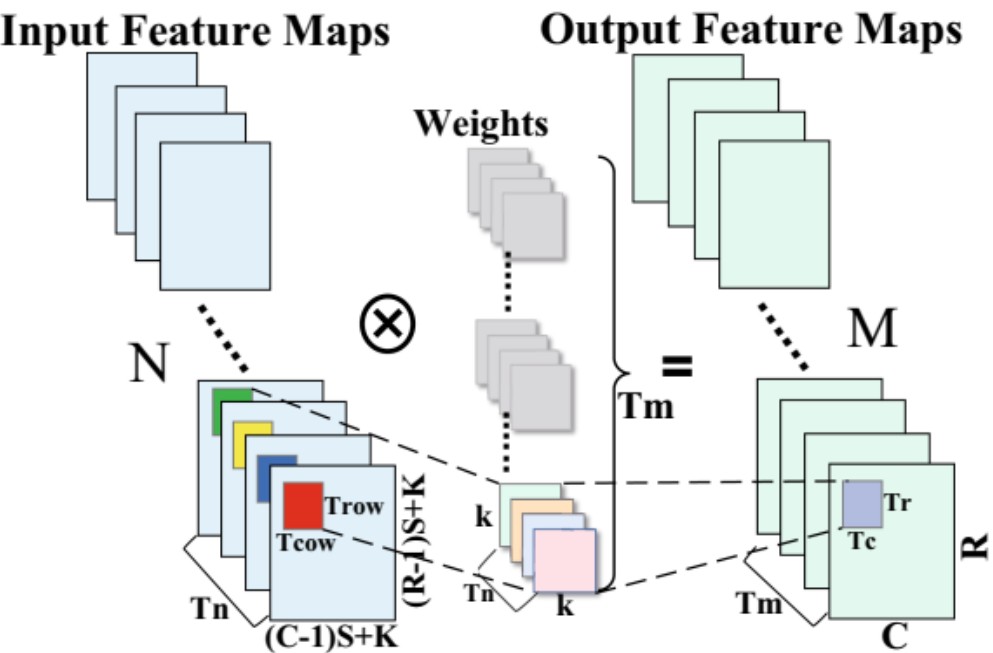

**Figure 8.** Calculation of parallel block.

### 4.3.2. Loop Tiling

The data locality computed by the convolution is used in the loop tiling. In data locality, entire data are divided as multiple smaller blocks, which are preserved in the on-chip buffers. The designed blocks are shown in Figure 8. From the DRAM, the pixel blocks of $T_{row} \times T_{cow}$ and respective $T_m \times T_n \times K^2$ weights of IFMs are acquired in this tiling process. Next, the weight parameters and pixel blocks of the feature map are used again on the chip. An external memory access is minimized, latency is reduced, and performance is improved by preserving the intermediate results in the on-chip cache. After obtaining the final output pixel blocks, the $T_r \times T_c$ pixel blocks of output feature maps are taken as output in this tiling process.

## 5. Results and Discussion

This section provides the performance and comparative analysis of the RCNN-SOW-CO architecture. The design of RCNN-SOW-CO is evaluated with three well-known CNNs, such as AlexNet, VGG16, and VGG19. The implemented design is evaluated in the Xilinx Zynq 7020 FPGA device. Here, the fixed-point data are utilized with 16-bit for input and output feature maps, 8-bits for CONV layer weight, 4-bits for fully connected (FC) layer weight, and 32-bits for partial addition. Here, the Vivado HLS 2019.1 is utilized for synthesizing the accelerator written in C++ into the register transfer level (RTL) design.

Subsequently, Vivado 2019.1 is used for compiling the RTL code into a bitstream. Moreover, the MNIST dataset [24] is used for testing the proposed RCNN-SOW-CO architecture.

### 5.1. Performance Analysis of RCNN-SOW-CO Architecture

The performance analysis of RCNN-SOW-CO architecture is analyzed by means of feature map size, weight size, sparseness of the IFM, weight parameter proportion, and FPGA performances. Here, the performances are analyzed for the CNN accelerator with SOW-CO and without SOW-CO.

The investigation of feature map size and weight size for RCNN-Alexnet with and without SOW-CO is shown in Table 2. Figures 9 and 10 show the graphical illustration of feature map size and weight size, respectively. From the table and figures, it is known that the feature map and weight sizes of RCNN-Alexnet with SOW-CO are less when compared to the RCNN without SOW-CO. The sizes of the feature map and weight are reduced in the RCNN with SOW-CO due to the parameter reduction achieved by using sparse matrix.

**Table 2.** Analysis of feature map and weight sizes for Alexnet.

| Alexnet Layers | Feature Map Size (Mb) | | Weight Size (Mb) | |
|:---:|:---:|:---:|:---:|:---:|
| | without SOW-CO | with SOW-CO | without SOW-CO | with SOW-CO |
| conv 1 | 16 | 14 | 3 | 2 |
| conv 2 | 13 | 10 | 5 | 5 |
| conv 3 | 11 | 9 | 8 | 7 |
| conv 4 | 9 | 7 | 12 | 9 |
| conv 5 | 7 | 6 | 14 | 12 |

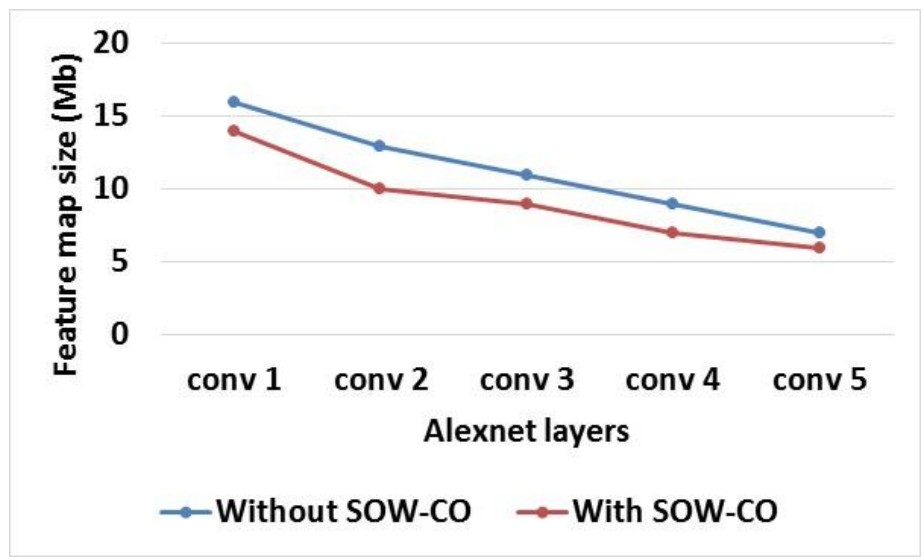

**Figure 9.** Graphical illustration of feature map size for Alexnet.

Table 3 shows the analysis of feature map and weight sizes for RCNN-VGG16 with and without SOW-CO. Additionally, the comparison of feature map and weight sizes for feature map size and weight size are shown in Figures 11 and 12, respectively. This analysis shows that the feature map and weight sizes of the RCNN with SOW-CO are less when compared to the RCNN without SOW-CO. The architecture without SOW-CO causes higher feature map and weight sizes, because it does not have optimized architecture for convolution, as well as the weight value not being optimized during the computation process.

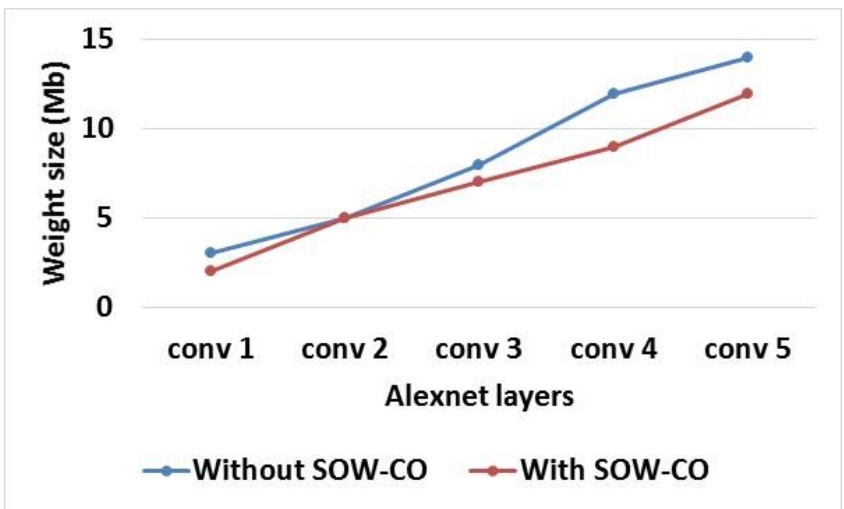

**Figure 10.** Graphical illustration of weight size for Alexnet.

**Table 3.** Analysis of feature map and weight sizes for VGG16.

| VGG16 Layers | Feature Map Size (Mb) | | Weight Size (Mb) | |
|---|---|---|---|---|
| | without SOW-CO | with SOW-CO | without SOW-CO | with SOW-CO |
| conv 1 | 21 | 19 | 4 | 2 |
| conv 2 | 18 | 17 | 4 | 3 |
| conv 3 | 18 | 16 | 7 | 5 |
| conv 4 | 15 | 14 | 8 | 6 |
| conv 5 | 12 | 10 | 10 | 8 |
| conv 6 | 14 | 10 | 13 | 10 |
| conv 7 | 12 | 9 | 14 | 12 |
| conv 8 | 10 | 7 | 16 | 15 |
| conv 9 | 8 | 7 | 20 | 17 |
| conv 10 | 6 | 5 | 21 | 18 |
| conv 11 | 5 | 3 | 21 | 20 |
| conv 12 | 4 | 2 | 24 | 23 |
| conv 13 | 3 | 2 | 25 | 23 |

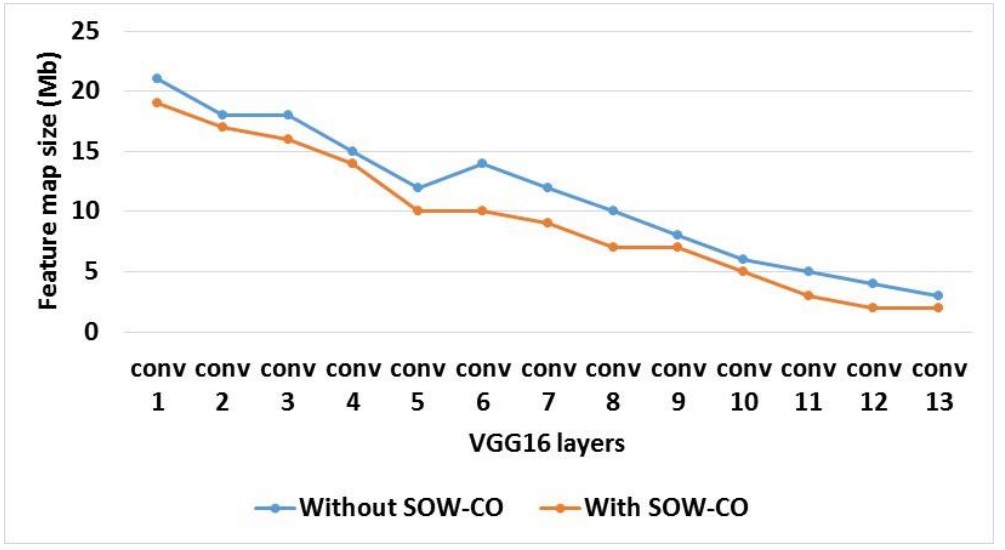

**Figure 11.** Graphical illustration of feature map size for VGG16.

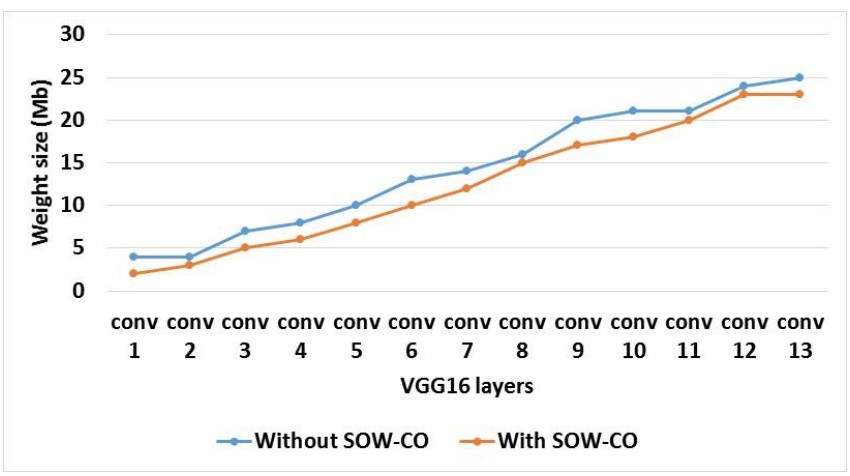

**Figure 12.** Graphical illustration of weight size for VGG16.

The investigation of feature map and weight sizes for RCNN-VGG19 with and without SOW-CO is shown in Table 4. Figures 13 and 14 show the graphical illustrations of feature map and weight sizes, respectively. From the analysis, it is known that the feature map and weight sizes of RCNN-VGG19 with SOW-CO are less than the RCNN without SOW-CO. The utilization of sparse matrix in weight and convolutional optimization reduces the sizes of the feature map and weight.

**Table 4.** Analysis of feature map and weight sizes for VGG19.

| VGG19 Layers | Feature Map Size (Mb) | | Weight Size (Mb) | |
|---|---|---|---|---|
| | without SOW-CO | with SOW-CO | without SOW-CO | with SOW-CO |
| conv 1 | 19 | 18 | 4 | 2 |
| conv 2 | 18 | 16 | 5 | 3 |
| conv 3 | 17 | 18 | 7 | 3 |
| conv 4 | 17 | 14 | 10 | 5 |
| conv 5 | 14 | 13 | 8 | 8 |
| conv 6 | 12 | 10 | 13 | 10 |
| conv 7 | 10 | 8 | 13 | 11 |
| conv 8 | 9 | 8 | 17 | 14 |
| conv 9 | 9 | 6 | 19 | 17 |
| conv 10 | 8 | 4 | 20 | 17 |
| conv 11 | 6 | 3 | 22 | 19 |
| conv 12 | 5 | 2 | 25 | 20 |
| conv 13 | 2 | 2 | 27 | 22 |

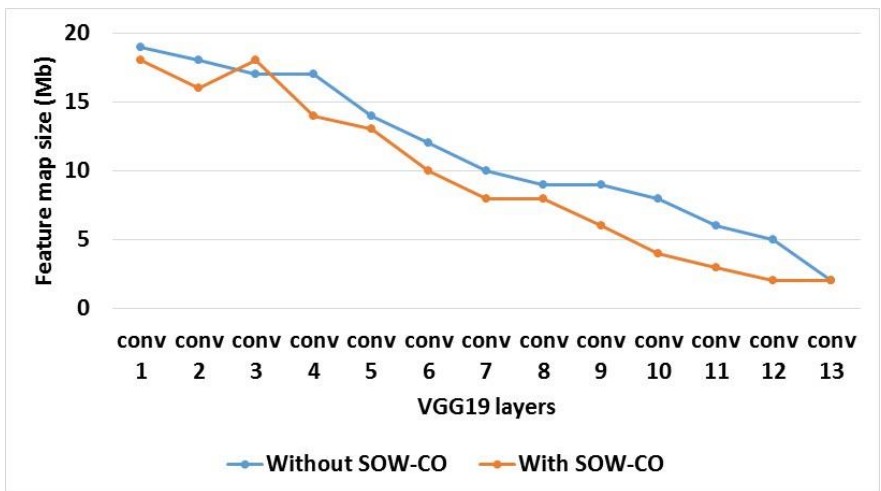

**Figure 13.** Graphical illustration of feature map size for VGG19.

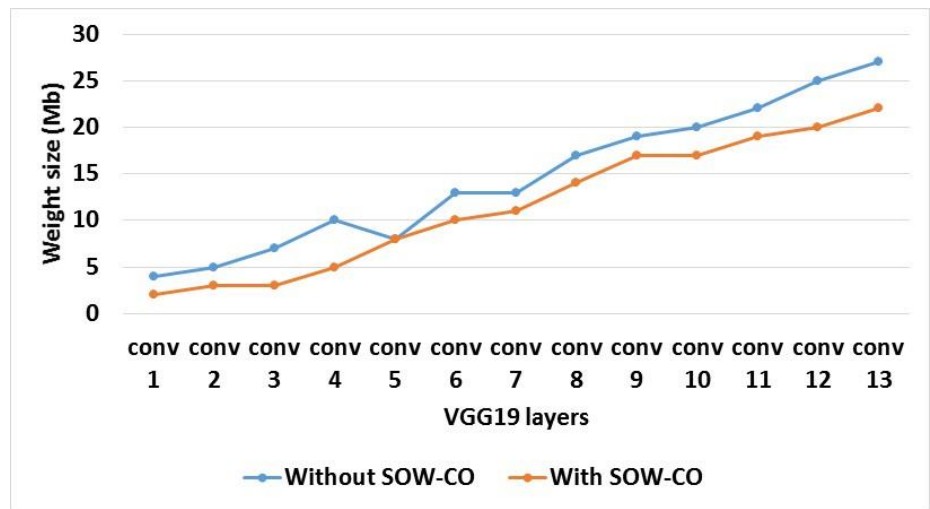

**Figure 14.** Graphical illustration of weight size for VGG19.

An evaluation of sparseness of the IFM and weight parameter proportion for RCNN-Alexnet with and without SOW-CO is shown in Table 5. The graphical illustrations of the IFM's sparseness and weight parameter proportion for RCNN-Alexnet are shown in Figures 15 and 16, respectively. This analysis shows that the IFM's sparseness and weight parameter proportion of RCNN-Alexnet with SOW-CO are less when compared to the RCNN-Alexnet without SOW-CO. The optimization of weight using the sparsity is used to optimize the IFM's sparseness and weight parameter.

**Table 5.** Analysis of sparseness of the IFM and weight parameter proportion for Alexnet.

| Alexnet Layers | Sparseness of the IFM | | Weight Parameter Proportion | |
|---|---|---|---|---|
| | without SOW-CO | with SOW-CO | without SOW-CO | with SOW-CO |
| conv 1 | 0.11 | 0.05 | 0.21 | 0.14 |
| conv 2 | 0.19 | 0.10 | 0.37 | 0.23 |
| conv 3 | 0.37 | 0.18 | 0.44 | 0.30 |
| conv 4 | 0.41 | 0.33 | 0.58 | 0.39 |
| conv 5 | 0.59 | 0.47 | 0.60 | 0.47 |
| fc1 | 0.65 | 0.51 | 0.83 | 0.61 |
| fc2 | 0.77 | 0.69 | 0.89 | 0.78 |
| fc3 | 0.90 | 0.82 | 0.93 | 0.88 |

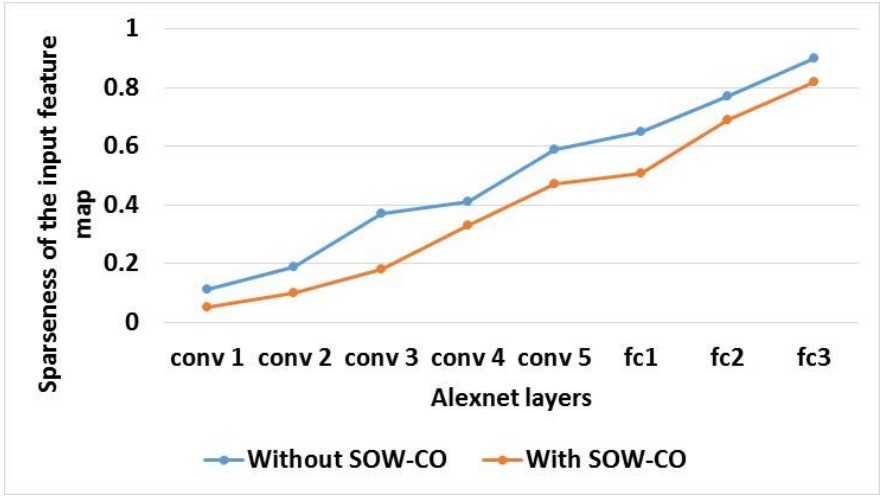

**Figure 15.** Graphical illustration of IFM's sparseness for Alexnet.

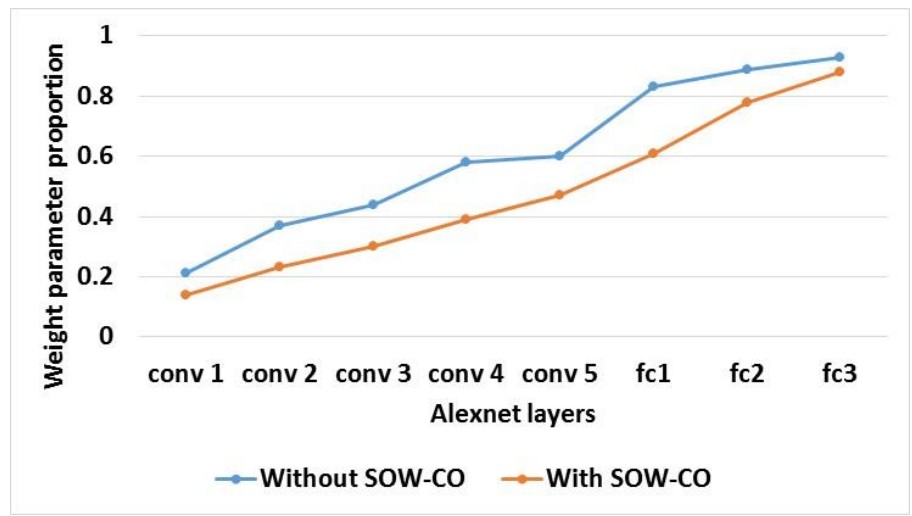

**Figure 16.** Graphical illustration of weight parameter proportion for Alexnet.

Table 6 shows the analysis of sparseness of the IFM and weight parameter proportion for RCNN-VGG16 with and without SOW-CO. Additionally, the comparison of feature map size and weight size for IFM's sparseness and weight parameter proportion are shown in Figures 17 and 18, respectively. This analysis shows that the IFM's sparseness and weight parameter proportion of the RCNN with SOW-CO are less when compared to the RCNN without SOW-CO. The IFM's sparseness and weight parameter proportion are increased because of the CNN without any convolutional and weight optimization.

**Table 6.** Analysis of sparseness of the IFM and weight parameter proportion for VGG16.

| VGG16 Layers | Sparseness of the IFM | | Weight Parameter Proportion | |
|:---:|:---:|:---:|:---:|:---:|
| | without SOW-CO | with SOW-CO | without SOW-CO | with SOW-CO |
| conv 1 | 0.09 | 0.04 | 0.11 | 0.08 |
| conv 2 | 0.12 | 0.10 | 0.18 | 0.12 |
| conv 3 | 0.15 | 0.12 | 0.27 | 0.19 |
| conv 4 | 0.24 | 0.19 | 0.33 | 0.27 |
| conv 5 | 0.29 | 0.24 | 0.39 | 0.31 |
| conv 6 | 0.31 | 0.25 | 0.46 | 0.38 |
| conv 7 | 0.33 | 0.29 | 0.50 | 0.44 |
| conv 8 | 0.41 | 0.37 | 0.51 | 0.46 |
| conv 9 | 0.47 | 0.40 | 0.59 | 0.52 |
| conv 10 | 0.54 | 0.44 | 0.67 | 0.55 |
| conv 11 | 0.61 | 0.53 | 0.72 | 0.67 |
| conv 12 | 0.66 | 0.60 | 0.75 | 0.70 |
| conv 13 | 0.72 | 0.67 | 0.77 | 0.70 |
| fc1 | 0.89 | 0.77 | 0.85 | 0.76 |
| fc2 | 0.93 | 0.80 | 0.90 | 0.83 |
| fc3 | 0.97 | 0.81 | 0.96 | 0.85 |

An evaluation of sparseness of the IFM and weight parameter proportion for RCNN-VGG19 with and without SOW-CO is shown in Table 7. The graphical illustrations of the IFM's sparseness and weight parameter proportion for RCNN-VGG19 are shown in Figures 19 and 20, respectively. This analysis shows that the IFM's sparseness and weight parameter proportion of RCNN-VGG19 with SOW-CO are less when compared to the RCNN-VGG19 without SOW-CO.

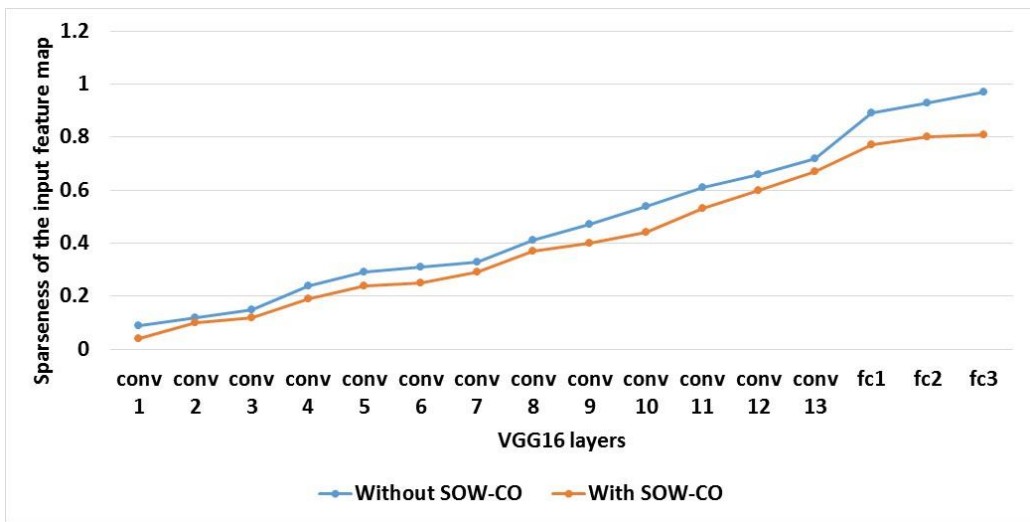

**Figure 17.** Graphical illustration of IFM's sparseness for VGG16.

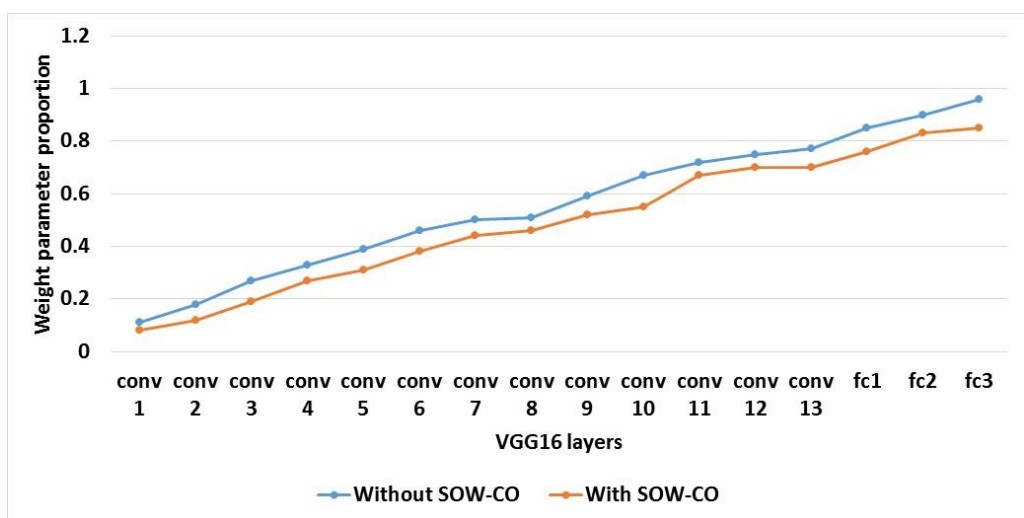

**Figure 18.** Graphical illustration of weight parameter proportion for VGG16.

**Table 7.** Analysis of sparseness of the IFM and weight parameter proportion for VGG19.

| VGG19 Layers | Sparseness of the IFM | | Weight Parameter Proportion | |
|---|---|---|---|---|
| | without SOW-CO | with SOW-CO | without SOW-CO | with SOW-CO |
| conv 1 | 0.12 | 0.08 | 0.10 | 0.07 |
| conv 2 | 0.17 | 0.11 | 0.14 | 0.09 |
| conv 3 | 0.20 | 0.17 | 0.16 | 0.10 |
| conv 4 | 0.24 | 0.21 | 0.24 | 0.16 |
| conv 5 | 0.29 | 0.24 | 0.29 | 0.21 |
| conv 6 | 0.33 | 0.28 | 0.35 | 0.27 |
| conv 7 | 0.38 | 0.33 | 0.39 | 0.33 |
| conv 8 | 0.47 | 0.39 | 0.43 | 0.38 |
| conv 9 | 0.55 | 0.46 | 0.48 | 0.41 |
| conv 10 | 0.64 | 0.58 | 0.51 | 0.47 |
| conv 11 | 0.68 | 0.61 | 0.55 | 0.49 |
| conv 12 | 0.85 | 0.67 | 0.64 | 0.59 |
| conv 13 | 0.89 | 0.74 | 0.77 | 0.63 |
| fc1 | 0.92 | 0.80 | 0.82 | 0.67 |
| fc2 | 0.94 | 0.81 | 0.89 | 0.72 |
| fc3 | 0.97 | 0.83 | 0.94 | 0.77 |

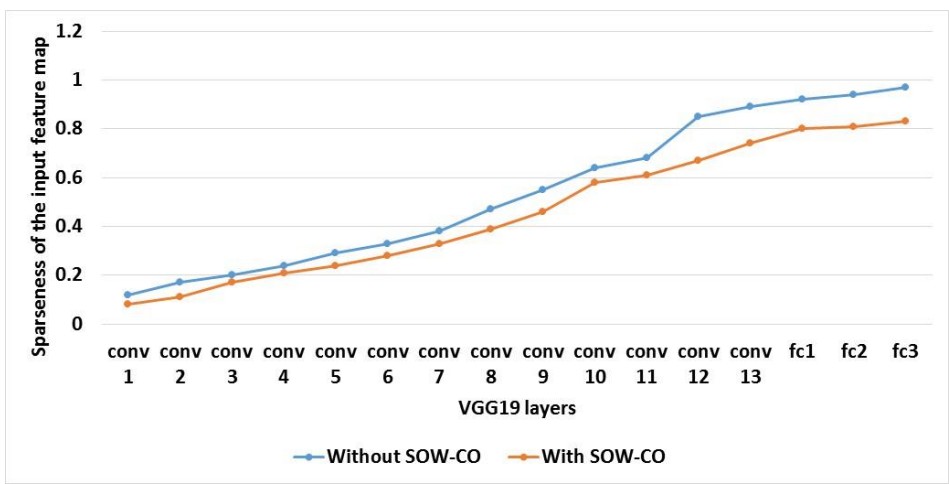

**Figure 19.** Graphical illustration of IFM's sparseness for VGG19.

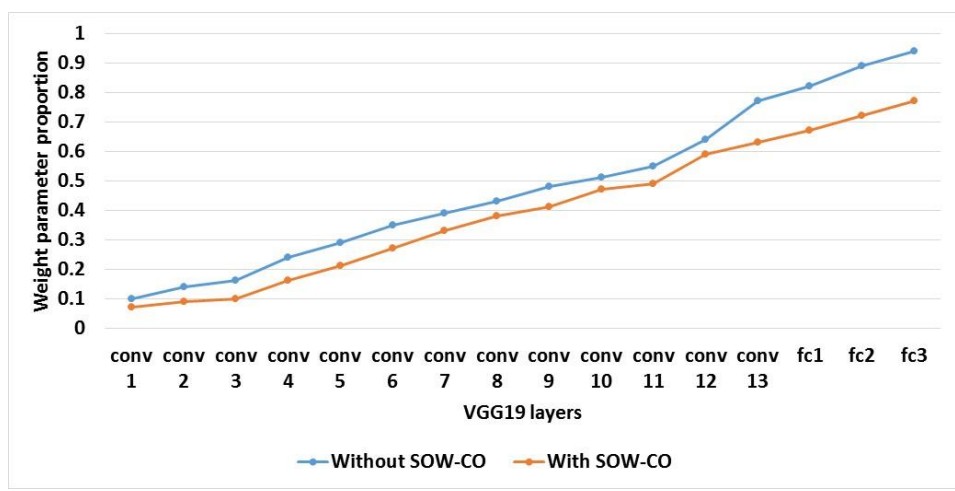

**Figure 20.** Graphical illustration of weight parameter proportion for VGG19.

The FPGA performances of RCNN are analyzed in terms of BRAM, DSP, LUT, slices, delay, power, and accuracy. Here, the performances are analyzed for the RCNN with and without SOW-CO architecture. Tables 8–10 show the analysis of FPGA performances for the AlexNet, VGG16, and VGG19, respectively. From the analysis, it is clear that the RCNN with SOW-CO architecture provides better performance than the RCNN without SOW-CO in terms of accuracy and execution time. The power consumption is minimized by using the data reuse accomplished in the RCNN with SOW-CO. Moreover, the loop unrolling used in the convolutional optimization helps to minimize the delay for the RCNN with SOW-CO.

**Table 8.** Analysis of FPGA performances for AlexNet.

| FPGA Performances | AlexNet | |
| --- | --- | --- |
| | without SOW-CO | with SOW-CO |
| BRAM | 54.3 | 40.1 |
| DSP | 210 | 157 |
| LUT | 6500 | 5150 |
| Slices | 450 | 300 |
| Delay(μs) | 90 | 50 |
| Power (W) | 5.67 | 1.01 |
| Accuracy (%) | 93.14 | 99.52 |
| Execution Time (ms) | 1.382 | 0.948 |

**Table 9.** Analysis of FPGA performances for VGG16.

| FPGA Performances | VGG16 | |
| --- | --- | --- |
| | **without SOW-CO** | **with SOW-CO** |
| BRAM | 57.01 | 43.14 |
| DSP | 220 | 160 |
| LUT | 6590 | 5180 |
| Slices | 500 | 310 |
| Delay($\mu$s) | 94 | 53 |
| Power (W) | 6.1 | 1.03 |
| Accuracy (%) | 92.11 | 99.41 |
| Execution Time (ms) | 0.927 | 0.756 |

**Table 10.** Analysis of FPGA performances for VGG19.

| FPGA Performances | VGG19 | |
| --- | --- | --- |
| | **without SOW-CO** | **with SOW-CO** |
| BRAM | 54.22 | 42.02 |
| DSP | 210 | 140 |
| LUT | 6410 | 5120 |
| Slices | 450 | 320 |
| Delay($\mu$s) | 85 | 49 |
| Power (W) | 5.94 | 1.00 |
| Accuracy (%) | 93.94 | 99.5 |
| Execution Time (ms) | 0.835 | 0.698 |

*5.2. Comparative Analysis*

The efficiency of the RCNN-SOW-CO architecture is evaluated by comparing it with recent CNN architectures. Existing architectures used to evaluate the RCNN-SOW-CO are OIDSCNN [17] and DPR-NN [23]. Here, the comparison is made with different FPGA devices, such as Zynq-7020 and Virtex-7. The comparative analyses of the Zynq-7020 and Virtex-7 for RCNN-SOW-CO architecture are shown in Tables 11 and 12, respectively. Figure 21 shows the comparison of BRAM for Zynq-7020. The proposed RCNN-SOW-CO is analyzed under various architecture layers, which are: AlexNet, VGG16, and VGG19. While considering the AlexNet layer, it attains BRAM performance of 40.1, 157 DSP, and 5150 LUTs. Meanwhile, the VGG16 layer achieves the BRAM performance of 43.14, 160 DSP, and 5180 LUTs. Then, finally, VGG19 attains 42.02 BRAM, 140 DSP, and 5120 LUTs. These details are clearly tabulated below in Table 11. This comparative analysis shows that the RCNN-SOW-CO architecture achieves a better performance than LP-CNN [20] under GoogLeNet case study for Virtex-7. A number of calculations in the RCNN are decreased by using the sparse matrix and convolutional optimization, which resulted in less hardware resources than the LP-CNN [20] for Virtex-7, which are tabulated in Table 12. Figure 22 shows the comparative analysis of power for Virtex-7. For the analysis, both the proposed RCNN-SOW-CO and existing LP-CNN [20] are processed with NVIDIA GPU. Alternatively, the spiking PE does not employ any DSPs, since there is no increase in spiking layers. Although, it takes additional hardware resources, such as LUTs. From the analysis, it is concluded that the proposed RCNN-SOW-CO with NVIDIA GPU achieves lower power consumption of 1.15 W when compared with the existing LP-CNN [20], which consumes 3.92 W.

**Table 11.** Comparative analysis of RCNN-SOW-CO architecture for Zynq-7020.

| Performances | OIDSCNN [17] | DPR-NN [23] | RCNN-SOW-CO | | |
| --- | --- | --- | --- | --- | --- |
| | | | **AlexNet** | **VGG16** | **VGG19** |
| BRAM | 25.5 | 58.5 | 40.1 | 43.14 | 42.02 |
| DSP | 219 | 167 | 157 | 160 | 140 |
| LUT | 7986 | 24980 | 5150 | 5180 | 5120 |

**Table 12.** Comparative analysis of RCNN-SOW-CO architecture for Virtex-7.

| Performances | LP-CNN [20] | RCNN-SOW-CO |
|---|---|---|
| BRAM | 1134 | 938.43 |
| LUT | 407,290 | 406,690 |
| Power (W) | 3.92 | 1.15 |

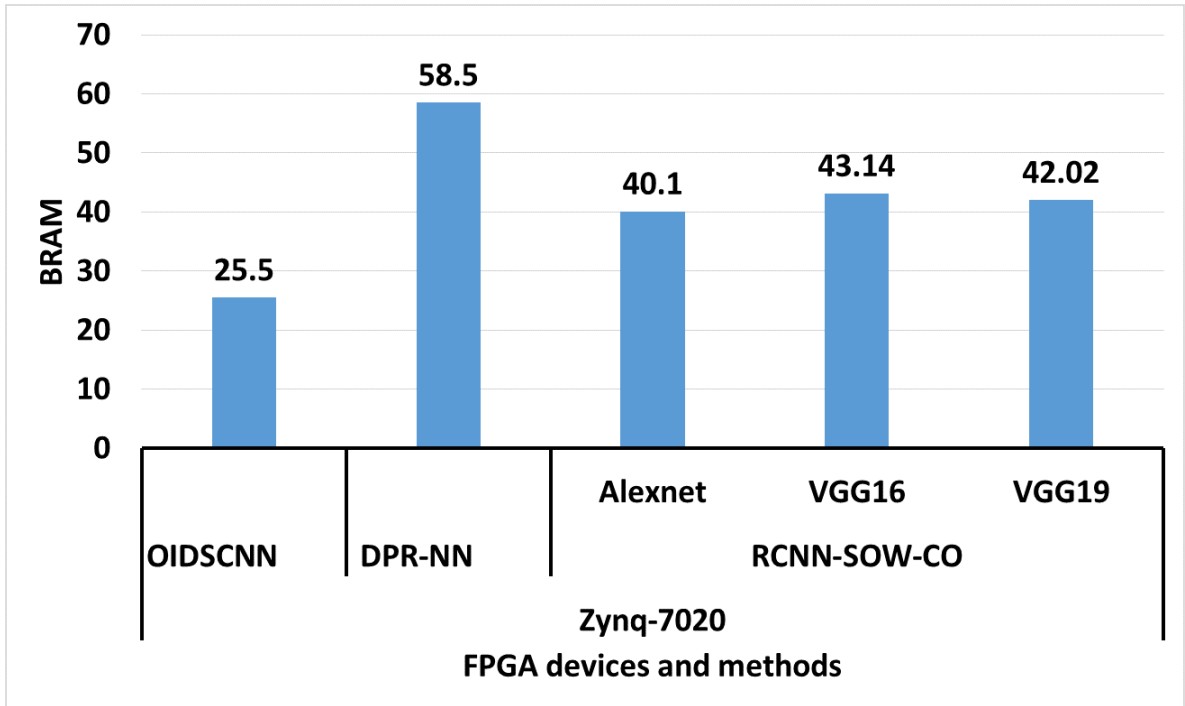

**Figure 21.** Comparison of BRAM. OIDSCNN [17], DPR-NN [23].

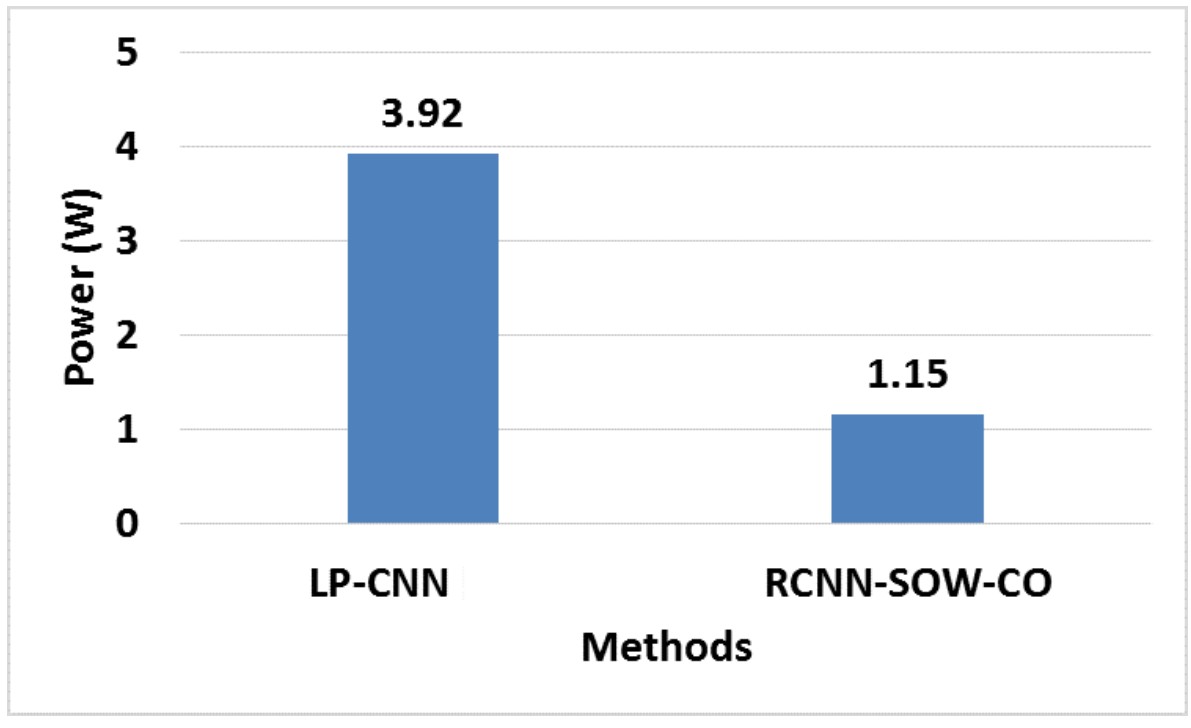

**Figure 22.** Comparison of power for Virtex-7. LP-CNN [20].

## 6. Conclusions

In this paper, the RCNN accelerator is proposed over the FPGA, along with weight optimization and convolutional optimization. The configuration instructions saved in the DDR are used to reconfigure the RCNN with the configuration register. Data access among the on-chip buffer and DDR are minimized by using the special functional layer, which includes a pooling, BN, and activation. The power utilized by this RCNN-SOW-CO is minimized in two ways; one is the reduction in on-chip data movement and the other one is a data reuse approach accomplished in the RCNN accelerator. Here, an number of computations used in the RCNN accelerator are minimized based on the matrix representation of sparse weight. Moreover, the loop unrolling used in the convolutional optimization is used to increase the RCNN accelerator's speed. From the results, it is concluded that the RCNN-SOW-CO provides the improved performances compared to the OIDSCNN, LP-CNN, and DPR-NN. The LUT of RCNN-SOW-CO with AlexNet designed in the Zynq-7020 is 5150, which is less than the OIDSCNN and DPR-NN. In the future, the proposed RCNN accelerator can be utilized to perform faster real-time object detection.

**Author Contributions:** The paper investigation, resources, data curation, writing—original draft preparation, writing—review and editing, and visualization were conducted by K.M.V.G. and S.M. The paper conceptualization and software were conducted by A.A. The validation, formal analysis, methodology, supervision, project administration, and funding acquisition of the version to be published were conducted by S.R. and P.B.D. All authors have read and agreed to the published version of the manuscript.

**Funding:** This research received no external funding.

**Data Availability Statement:** The data presented in this study are openly available in MNIST dataset at doi: 10.1109/MSP.2012.2211477, reference number [24].

**Conflicts of Interest:** The authors declare no conflict of interest.

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
