# Peer review of "FPGA-Based Reconfigurable Convolutional Neural Network Accelerator Using Sparse and Convolutional Optimization"

_electronics, doi:10.3390/electronics11101653_

Round 1

Reviewer 1 Report

1. Figure 1 is unclear. The author needs to explain Figure 1. Data flow-based reconfigurable architecture more clearly.

2. There are some issues with the statement “The outputs are calculated into special function buffer at the PE array streams.” in lines 151 and 152, which are difficult to understand.

3. How are the Static Random Access Memory (SRAM) banks connected in parallel at the special function buffer? A detailed description needs to be added here.

4. The PE architecture diagram is inconsistent with the text description of “the reconfigurable PE has nine multipliers” in this article. The author needs to revise this part of the text statement or architecture diagram.

5. A error exists in Table 1. Usage of peak multiplier for various kernel sizes. The author needs to revise it. The author should check all the Tables and Figures.

Author Response

Comments and Suggestions for Authors 1

  1. Figure 1 is unclear. The author needs to explain Figure 1. Data flow-based reconfigurable architecture more clearly.

Answer:

Thank you for your useful comment. As per the reviewer’s comment, we have improved the quality of figure 1. Furthermore, we have presented a detailed explanation for Data flow-based reconfigurable architecture at the beginning of section 4.

  1. There are some issues with the statement “The outputs are calculated into special function buffer at the PE array streams.” in lines 151 and 152, which are difficult to understand.

Answer:

Thank you for your important comment. As per the reviewer’s comment, we have provided the proper statement in lines 151 and 152. Now the reader can understand the exact meaning of those lines.

  1. How are the Static Random Access Memory (SRAM) banks connected in parallel at the special function buffer? A detailed description needs to be added here.

Answer:

Thank you for your useful comment. As per the reviewer’s comment, we have included a detailed description about the Static Random Access Memory (SRAM) connections at the beginning of section 4.

  1. The PE architecture diagram is inconsistent with the text description of “the reconfigurable PE has nine multipliers” in this article. The author needs to revise this part of the text statement or architecture diagram.

Answer:

Thank you for your important comment. As per the reviewer’s comment, we have modified the text statement in PE architecture diagram at section 4.1.

  1. A error exists in Table 1. Usage of peak multiplier for various kernel sizes. The author needs to revise it. The author should check all the Tables and Figures.

Answer:

Thank you for your important comment. As per the reviewer’s comment, we have modified the table 1. Also, we have verified and updated all the 

Reviewer 2 Report

This paper presents the Sparse Optimization of Weight (SOW) and Convolutional Optimization (CO) techniques to improve the performance of a Reconfigurable Convolutional Neural Network (RCNN) accelerator implemented in FPGAs.

The main goal of the proposed optimizations (SOW and CO) is to optimize the feature map and weight sizes of the RCNN accelerator, so that the overall hardware resource usages and resulting power consumption can be minimized in FPGA.

The proposed optimization techniques are evaluated in terms of feature map size, weight size, sparseness of the Input Feature Map, weight parameter proportion, BRAM, DSP, LUTs, delay, power, and accuracy.

The evaluation section also compares the proposed optimized RCNN (RCNN-SOW-CO) against the existing CNN architectures (OIDSCNN, LP-CNN, and DPR-NN).

This paper nicely explains the base architecture of the proposed RCNN and high-level overview of the proposed optimizations.

The evaluation results seem to provide enough details demonstrating the benefits of the proposed optimizations in terms of the hardware resource usage and power consumption.

However, this paper still has several rooms for further improvements.

First, it is unclear of the core contributions of this paper; the proposed optimizations (SOW and CO) are not new, used in many of the existing FPGA CNN implementations, and this work may not be the first one that applies SOW and CO techniques to RCNN.

Therefore, this paper may not have enough contributions in terms of novelties.

The main contributions of this paper may be in evaluating the effect of the SOW and CO techniques on RCNN.

However, there may not be enough new contributions in evaluating the performance of SOW and CO techniques on RCNN since there exist many previous work that evaluated these techniques on the same or similar CNN architectures.

Second, the evaluation section can be improved further. 

The evaluation section compares the hardware resource usage of the proposed RCNN-SOW-CO against other existing CNN architectures (OIDSCNN, LP-CNN, DPR-NN), but the naive comparison against those existing CNN work may not be useful enough, since each of them targets different problems.

The related work comparison mainly focuses on the resource usage and power consumption, but it will be also important to compare the impact of the proposed work in terms of execution performance and accuracy.

The comparison between RCNN-SOW-CO and RCNN shows that RCNN-SOW-CO can improve the accuracy in all tested CNNs (AlexNet, VGG16, and VGG19), but this paper does not provide enough explanations on this unexpected behaviors.

Therefore, it will be better if this paper clarifies the core contributions and beefs up the evaluation sections.

Author Response

Comments and Suggestions for Authors 2

  1. First, it is unclear of the core contributions of this paper; the proposed optimizations (SOW and CO) are not new, used in many of the existing FPGA CNN implementations, and this work may not be the first one that applies SOW and CO techniques to RCNN. Therefore, this paper may not have enough contributions in terms of novelties.

Answer:

Thank you for your important comment. As per the reviewer’s comment, we have provided the core contribution of this research at the end of section 1.

  1. The main contributions of this paper may be in evaluating the effect of the SOW and CO techniques on RCNN. However, there may not be enough new contributions in evaluating the performance of SOW and CO techniques on RCNN since there exist many previous works that evaluated these techniques on the same or similar CNN architectures.

Answer:

Thank you for your useful comment. As per the reviewer’s comment, we have included the novelty of this research by combining SOW and CO techniques on RCNN at the end of introduction part.

Second, the evaluation section can be improved further. 

  1. The evaluation section compares the hardware resource usage of the proposed RCNN-SOW-CO against other existing CNN architectures (OIDSCNN, LP-CNN, DPR-NN), but the naive comparison against those existing CNN work may not be useful enough, since each of them targets different problems.

Answer:

Thank you for your valuable comment. As per the reviewer’s comment, we have modified and unified the comparison target of Table 11 and 12 at section 5.2.

  1. The related work comparison mainly focuses on the resource usage and power consumption, but it will be also important to compare the impact of the proposed work in terms of execution performance and accuracy.

Answer:

Thank you for your useful comment. As per the reviewer’s comment, we have included the impact of the proposed work in terms of execution performance and accuracy (Refer Table 8,9 and 10) at the end of section 5.1.

  1. The comparison between RCNN-SOW-CO and RCNN shows that RCNN-SOW-CO can improve the accuracy in all tested CNNs (AlexNet, VGG16, and VGG19), but this paper does not provide enough explanations on this unexpected behaviors. Therefore, it will be better if this paper clarifies the core contributions and beefs up the evaluation sections.

Answer:

Thank you for your valuable comment. As per the reviewer’s comment, we have described the accuracy comparative performances during AlexNet, VGG16, and VGG19 cases. Furthermore, we have clarified the core contribution

Reviewer 3 Report

The authors proposed an algorithm for optimisation of accelerating CNN using a FPGA structure.  The paper organised on 6 chapters starting with an introduction about the subject and followed by existing works in the field of paper title and after is exposed the problem statement.  The solution is presented in chapter 4 beginning with proposed architecture and sustained by result and discussions in chapter 5. There is presented also the comparation with another references.

I suggest to improve the article trough comparation with dedicated system already used at large scale (for. Example NVIDIA GPU) and measured the consumption for the same quantity  of information processed

Author Response

Comments and Suggestions for Authors 3

The authors proposed an algorithm for optimisation of accelerating CNN using a FPGA structure.  The paper organised on 6 chapters starting with an introduction about the subject and followed by existing works in the field of paper title and after is exposed the problem statement.  The solution is presented in chapter 4 beginning with proposed architecture and sustained by result and discussions in chapter 5. There is presented also the comparation with another references.

  1. I suggest to improve the article trough comparation with dedicated system already used at large scale (for. Example NVIDIA GPU) and measured the consumption for the same quantity of information processed

Answer:

Thank you for your valuable comment. As per the reviewer’s comment, we have analysed the proposed method with NVIDIA GPU for the same quantity of information which is tabulated (Table 11) at section 5.2.

Round 2

Reviewer 2 Report

This is my second review of this paper. 

The revised version seems to address most of concerns raised by reviewers.

However, there exist a few remaining issues in the revised version:

Figure 22 compares the power consumption of LP-CNN and RCNN-SOW-CO on Virtex-7, but the numbers are different from those in Table 12. 

Section 5.2 includes statements explaining the Figure 22 at the end, but the actual explanations are about power consumption comparison between RCNN-SOW-CO and DPR-NN with yet-another power consumption numbers (1288 mW for RCNN-SOW-CO and 1528 mW for DPR-NN); Figure 22 and explanations in Section 5.2 do no match! 

Please correct either Figure 22 or the section 5.2 (or both) to be consistent with each other.

The modified texts have several minor grammatical errors (e.g., section 4); please correct those errors too.

Author Response

Reviewer 2:

Comments and Suggestions for Authors

This is my second review of this paper. 

The revised version seems to address most of concerns raised by reviewers.

However, there exist a few remaining issues in the revised version:

Figure 22 compares the power consumption of LP-CNN and RCNN-SOW-CO on Virtex-7, but the numbers are different from those in Table 12. 

Answer:

Thanks for your observation and valuable time. Based on your comment, we have verified and updated the Figure 22 according to the Table 12.

Section 5.2 includes statements explaining the Figure 22 at the end, but the actual explanations are about power consumption comparison between RCNN-SOW-CO and DPR-NN with yet-another power consumption numbers (1288 mW for RCNN-SOW-CO and 1528 mW for DPR-NN); Figure 22 and explanations in Section 5.2 do no match! 

Please correct either Figure 22 or the section 5.2 (or both) to be consistent with each other.

Answer:

Thanks for your observation and valuable time. As per your comment, we have verified and updated the section 5.2.

The modified texts have several minor grammatical errors (e.g., section 4); please correct those errors too.

Answer:

Thank you for your useful comments. Based on your comment, we have corrected the grammatical errors in the updated manuscript.
